# Training and Technology Acceptance of ChatGPT in University Students of Social Sciences: A Netcoincidental Analysis

**DOI:** 10.3390/bs14070612

**Published:** 2024-07-18

**Authors:** Elena María García-Alonso, Ana Cristina León-Mejía, Roberto Sánchez-Cabrero, Raquel Guzmán-Ordaz

**Affiliations:** 1Department of Sociology and Communication, Faculty of Social Sciences, University of Salamanca, 37007 Salamanca, Spain; aleon@usal.es (A.C.L.-M.); r.guzman@usal.es (R.G.-O.); 2Department of Evolutionary Psychology and Education, Faculty of Teacher Training and Education, Autonomous University of Madrid, 28049 Madrid, Spain; roberto.sanchez@uam.es

**Keywords:** ChatGPT, education, reticular analysis of coincidences, networks, technology acceptance model, training

## Abstract

This study analyzes the perception and usage of ChatGPT based on the technology acceptance model (TAM). Conducting reticular analysis of coincidences (RAC) on a convenience survey among university students in the social sciences, this research delves into the perception and utilization of this artificial intelligence tool. The analysis considers variables such as gender, academic year, prior experience with ChatGPT, and the training provided by university faculty. The networks created with the statistical tool “CARING” highlight the role of perceived utility, credibility, and prior experience in shaping attitudes and behaviors toward this emerging technology. Previous experience, familiarity with video games, and programming knowledge were related to more favorable attitudes towards ChatGPT. Students who received specific training showed lower confidence in the tool. These findings underscore the importance of implementing training strategies that raise awareness among students about both the potential strengths and weaknesses of artificial intelligence in educational contexts.

## 1. Introduction

Artificial intelligence (AI) represents the generational technological leap of our time [1,2,3]. It is not a new phenomenon, since the first developments were made more than seven decades ago [4]. However, we can consider that the last five years (2020–2024) have been the period representing a definitive explosion in the area. A significant part of these advances can be attributed to the commercial launch of the revolutionary tool Chat Generative Pre-trained Transformer, commonly known as ChatGPT, which has shown the world the enormous potential of AI, much of which is still being explored.

ChatGPT is a natural language processing (NLP) system developed by the company OpenAI. It relies on large language models (LLM) that are pre-trained using deep neural networks to process extensive volumes of data. This technology equips ChatGPT with the ability to learn linguistic and contextual patterns, which is essential for understanding and generating responses to open-ended prompts. Additionally, by emulating human cognitive processes, ChatGPT interacts with users, enabling dynamic conversations [5,6,7,8,9]. 

Since its launch in 2022, building on previous beta versions, numerous studies have addressed the social implications of ChatGPT at all levels. This has resulted in many arguments in favor of its use in various fields, as well as numerous arguments against it. Currently, we are at a historical moment where more research is necessary to determine the social implications of using this tool in essential areas, such as the educational field. The controversy over its use is intense, and technological advances face many ethical conflicts that are difficult to resolve. This study is framed within this context, aiming to shed light on the issue by analyzing the perception and usage of ChatGPT. This research is based on the technology acceptance model (TAM) and employs reticular analysis of coincidences (RAC) among university students in the social sciences.

## 2. Literature Review

### 2.1. ChatGPT and the Technology Acceptance Model (TAM)

The technology acceptance model (TAM) is a theoretical model that aims to explain and predict user acceptance and the usage of technology [10,11]. It has become one of the most widely used models for studying the adoption of information technologies [12]. The TAM focuses on two main factors: (1) Perceived usefulness, i.e., the degree to which a person believes that using a particular technology will enhance their job performance or how they perceive benefits in specific tasks, and (2) Perceived ease of use, or the degree to which a person believes that using a particular technology will be free of effort. This includes the intuitiveness of the user interface, the simplicity of issuing commands, and the clarity of the responses provided by the tool. Ease of use can also encompass the initial learning curve and the ongoing ease of use in various applications.

Based on the psychology-driven theories of reasoned action (TRA) and planned behavior (TPB), the TAM has become a key framework in understanding user behaviors regarding technology [13]. Subsequent developments of the TAM have introduced additional factors that may influence technology acceptance. For instance, the attitude towards using captures users’ overall opinions and feelings about the technology. Behavioral intention of use and actual use focus on users’ plans to use the technology and their actual usage patterns. Social influence examines how the opinions of others or organizational culture impact technology acceptance. Facilitating conditions pertain to the perceived availability of resources and support necessary for using the technology. By applying the TAM, researchers and developers can identify factors that promote or hinder the acceptance of ChatGPT. Elements such as training, support, and experience with similar technologies have become the focus of recent attention because they can boost users’ confidence and perceived usefulness, reduce resistance to adopting technology, and lower perceived complexity, to mention just a few examples.

The extension of the TAM with other theoretical perspectives aiming to explain and predict users’ intentions to use information technology gives rise to the unified theory of acceptance and use of technology (UTAUT) [14,15,16]. This approach identifies four key constructs influencing intention and usage behavior, namely, performance expectancy, which is the degree to which an individual believes that using the technology will enhance their job performance, effort expectancy, which is the degree of ease associated with the use of the technology, social influence, which is the extent to which an individual perceives that important others believe they should use the new technology, and facilitating conditions, which is the perception of the availability of resources and support necessary to use the technology. According to UTAUT, performance expectancy, effort expectancy, and social influence each affect users’ behavioral intention to use technology, whereas behavioral intention and facilitating conditions determine the actual use of the technology. 

The relationship between the TAM and UTAUT is evident in several aspects. Perceived usefulness in the TAM is akin to performance expectancy in the UTAUT, as both pertain to the belief that technology use will improve performance. Similarly, perceived ease of use in the TAM is equivalent to effort expectancy in the UTAUT, referring to the ease of technology use. While the TAM indirectly addresses social influence, the UTAUT explicitly includes it as a core construct. The UTAUT introduces facilitating conditions which are not explicitly present in TAM but are somewhat implied in the concept of ease of use. Both models focus on predicting the intention to use and actual use of technology.

### 2.2. The Emergence of ChatGPT in Educational Settings

ChatGPT finds diverse applications in education, spanning from generating and translating content to summarization in different formats, like stories, essays, letters, and tweets [17]. It is also used for creative tasks, such as composing music, crafting theatrical scripts, screenplay writing, and beyond. Moreover, ChatGPT excels in answering a spectrum of questions, delivering customized responses in various tones, including formal, informal, motivational, commercial, or academic. This adaptability makes it a valuable asset for language acquisition, enhancing writing proficiency, and synthesizing information across diverse research domains [18,19,20,21,22,23,24]. However, it has also been suggested that employing ChatGPT for writing purposes makes it necessary to teach more advanced skills reflecting students’ critical, analytical, and argumentative abilities [25,26].

Beyond its academic applications, ChatGPT holds the potential to enhance student motivation by aiding in task and class management, especially in online teaching scenarios and blended learning environments [23,27,28,29,30,31,32]. It also improves virtual and visual learning in health-related studies [33,34,35,36,37,38]. However, in some tasks related to medical education, ChatGPT does not always provide better learning outcomes than traditional methods [39,40]. 

Benefits of using it include assisting in finding pertinent resources and offering comprehensive feedback [41], covering aspects, such as content, structure, grammar, and spelling [23,27,28,42,43]. Moreover, it fosters collaboration among students [27,44,45] and enhances creativity, problem-solving, and critical thinking, and improves several metacognitive abilities [46,47,48,49]. It also increases accessibility and inclusion [20], extending its benefits to students with disabilities or those who may not be proficient in the native language they study or work in [21,45]. In summary, it functions as a customized tool fostering autonomy and inclusivity, thereby enhancing the overall learning experience [50].

However, the integration of ChatGPT in educational environments not only demonstrates a positive reception [20,42,48,51,52,53,54], as reflected in the growing number of scientific publications [55], but also raises numerous concerns and reservations related to ethics, academics, and the training of teachers to master this tool [41,56,57,58,59,60,61,62]. Concerns have been highlighted regarding the accuracy and generation of incorrect responses, which can be attributed to the quality, diversity, and complexity of the training data, as well as the input provided by users [7,57]. Further studies also underscore biases and unsuitable content arising from language models and algorithms, coupled with challenges in comprehending the nuances of human language [63,64].

Moreover, the scholarly discussion surrounding ChatGPT within the more critical framework of education emphasizes issues like plagiarism and ethical considerations [27,65,66,67,68], excessive dependence on technology [7,60,61,66], potential threats to critical thinking [7,45,60,69], concerns about data privacy and security [54,70,71], lack of human moderation in user interactions [72] and issues related to the digital divide [7,45]. 

### 2.3. Factors Influencing Technology Acceptance among University Students

The initial studies on the acceptance of ChatGPT soon applied the TAM (technology acceptance model) [72,73,74,75,76,77,78,79,80], showing that perceived usefulness and ease of use positively impacted attitude and intention to use ChatGPT, and that most students had a positive view of ChatGPT, finding it user-friendly and beneficial for completing assignments. Since then, numerous factors have been found to influence university students’ acceptance of ChatGPT, including the aforementioned perceived ease of use and perceived usefulness, social influence (i.e., peer recommendations), organizational support, self-evaluation judgments, information quality, reliability, experience, performance expectancy, hedonic motivation, price value, and habits and facilitating conditions [81,82,83,84,85,86,87,88,89,90,91,92,93,94,95,96,97]. The acceptance of ChatGPT by university students may be influenced by experience, habit, and behavioral intention, with facilitating conditions and user behavior also playing significant roles in technology adoption [98]. External factors such as stress and anxiety can negatively impact ChatGPT acceptance by diminishing motivation and perception about utility and usage [99,100,101]. However, in some cases, the stress leading to anxiety drives students to adopt this technology to meet deadlines, revealing a complex relationship between these psychological variables [101].

Research indicates that university students are increasingly utilizing AI tools like ChatGPT for academic assistance, with positive perceived results on performance, particularly in understanding complex concepts, accessing relevant study materials [102], and solving tasks with a higher degree of quality [60,97]. They particularly value that ChatGPT helps them increase self-efficacy while reducing mental effort, although students’ perceptions and interpretations may not always align with reality [97]. The benefits of using ChatGPT include facilitating adaptive learning, providing personalized feedback, supporting research, writing, and data analysis, and aiding in developing innovative assessments [103].

When examining studies that investigate moderating effects, a complex view emerges. Regarding the intention to use ChatGPT, it seems that higher levels of personal innovation may lead to stronger perceptions of usefulness, ease of use, information quality, and reliability, whereas lower levels make perceived risk more salient [84]. User acceptance, mediated by information quality, system quality, perceived learning value, and perceived satisfaction, plays an important role in determining users’ acceptance of ChatGPT [103,104]. According to the diffusion of innovation theory, it has been found that compatibility, observability, and trialability influence students’ adoption of ChatGPT in higher education. These factors are responsible for perceiving this tool as innovative, compatible, user-friendly, and useful [105]. Additionally, perceived ease of use may not directly predict learners’ attitudes but does so through the full mediator perceived usefulness. Those with positive attitudes toward the usefulness of ChatGPT have a higher level of behavioral intention, which positively and strongly predicts their actual use [75].

Regarding factors that could negatively impact the acceptance of ChatGPT among university students, research has found that only a minority perceive this tool as unsuitable for educational tasks. This group is concerned with its impact or bad performance on research, data analysis skills, and creative writing [59,93,106]. However, academic misconduct and plagiarism are the factors that worry students the most, with higher willingness to use it when the perceived risk of detecting the use of ChatGPT is low [102,106,107].

The studies mentioned earlier did not focus on the role of training on students’ perceptions of ChatGPT according to the technology acceptance model. Consequently, our main objective is to fill this gap in the emerging literature related to the educational use of ChatGPT. To accomplish this, we adapted Yilmaz’s questionnaire [72] based on an extended TAM to the Spanish context and included specific questions about prior training for undergraduate students in the faculties of social sciences and law. 

We conducted a comparative analysis of attitudes and perceptions regarding the utility, credibility, social influence, privacy and security, ease of use, and intention to use ChatGPT based on the type of training received for classroom use. The main hypothesis of our study is that prior training will have a decisive influence on students’ attitudes towards ChatGPT, breaking down prejudices, fears, and taboos regarding its academic and personal use.

## 3. Materials and Methods

### 3.1. Design and Procedures 

The study employed a cross-sectional, ex post facto design and utilized a convenience sample comprising a total of 216 Spanish students (72.3% female, 25.8% male, 1.9% preferred not to disclose this information), aged between 17 and 60 years: 17–19 (70%), 20–22 (22%), and 23–60 (7.7%). This latter percentage accounts for 4 sociology students in their third and fourth years, aged 33, 34, 48, and 60. In terms of disciplines, criminology (from the University of Salamanca, USAL) had the highest number of students, followed by sociology (USAL) and social education (University of Valladolid, UVA). This information is gathered in Table 1.

A survey evaluating the usage and perception of ChatGPT among social sciences students was conducted. The survey incorporated a previously validated questionnaire, along with additional sociodemographic, artificial intelligence, and technology usage questions. Data collection took place during October and November 2023 within class hours. It is important to note that there might be a bias towards students who regularly attend classes (and achieve higher grades) compared to those who do not. As per the survey results, 70% achieved notable grades, 17% outstanding, 8.3% passed, and only two students failed the previous university course (0.9%). The data collection method used did not allow for the segregation of students based on their degree.

In full compliance with the Research Ethics Committee Regulations of the University of Salamanca and the University of Valladolid, participation was voluntary and required obtaining informed written consent. This research was conducted in accordance with the regulations of the Research Ethics Committee of the Autonomous University of Madrid, which states in its protocol (Article 1.2) that questionnaires are outside the scope of application about specific assessments. The data were anonymized and securely stored for evaluation purposes. According to the Research Ethics Committee Regulations of the University of Salamanca and the University of Valladolid, students’ academic work produced in didactic activities as part of the curriculum may be utilized for research with their explicit written consent. This study adheres to a non-interventional approach, ensuring participant anonymity, in accordance with the Spanish Organic Law 3/2018, dated 5 December, on Data Protection and Guarantee of Digital Rights.

### 3.2. Instrument 

We used the questionnaire on attitudes toward ChatGPT, originally developed in English by Yilmaz [72]. It consists of 21 items distributed across seven dimensions: (1) perceived utility, (2) attitudes towards the use of ChatGPT, (3) perceived credibility, (4) perceived social influence, (5) perceived privacy and security, (6) perceived ease of use, and (7) behavioral intention to use ChatGPT. Table 2 below describes the seven dimensions of the instrument in detail.

In its adaptation to Spanish, the questionnaire underwent a direct and reverse translation process, followed by a final review by two experts. Due to the questionnaire’s simplicity, the Spanish version did not require substantial linguistic adjustments for comprehensibility. Confirmatory factor analysis (CFA) demonstrated a favorable model fit, reliability, and validity of the questionnaire (α = 0.855), and the results showed an overall positive perception of ChatGPT among the participants. Each item was measured on a Likert scale with the characteristics given in Table 3.

The survey was administered through the educational platforms of the University of Salamanca, “Studium”, and the Virtual Campus of the University of Valladolid. Participants completed the questionnaire during class hours under the supervision of the instructor. Responses submitted outside of class hours were excluded to ensure sample control. The independent variables included to explore variations in subjectivity concerning ChatGPT use encompassed gender, the received training type, academic performance, determined by the average grade of the prior academic course, and, as an indicator of digital skills, proficiency in programming and video game usage.

The data analysis proceeded in two phases. In the descriptive exploration of data, mean comparisons were computed considering both student characteristics and dimensions of the instrument. For inferential analyses, students were categorized based on their subjective evaluation of ChatGPT.

Subsequently, reticular coincidence analysis [108] was employed to uncover coincidences. This method is used in statistics and data analysis to identify and analyze patterns of coincidences or correlations among multiple variables in a visual and intuitive way. In the reticular structure created, the nodes represent variables, while the edges (lines connecting nodes) represent statistically significant coincidences or correlations between these nodes. Nodes can represent individual variables, events, or data points and are depicted as dots in the network. The edges illustrate the relationships or correlations between the nodes, indicating a significant relationship if two nodes are connected by an edge. The reticular graphs highlight only statistically significant matches, determined by Haberman residuals (*p* < 0.05). This means that the connections (edges) shown in the network are not random; they are statistically meaningful. The size of each node denotes the response percentage or the importance of that variable, with larger nodes indicating higher response rates or greater significance. On the other hand, the thickness of the lines indicates the strength of the match or relationship, as measured by the Haberman coefficient (thicker lines indicate stronger relationships, while thinner lines indicate weaker relationships).

We used the Caring tool (Proyect NetCoin. https://caring.usal.es/) to create the five networks presented in this study. This tool allowed for a detailed visualization of the statistically significant relationships identified in our analysis.

## 4. Results

The student demographic commonly found in social science classrooms, which was primarily women (as shown in Table 1), did not manifest a specific inclination toward technological topics. Notably, 13% admitted to not engaging in video games, and 31% possessed only basic knowledge, with a mere 20.9% displaying advanced proficiency in video game usage. Concerning programming knowledge, the majority claimed to have no understanding (60%), while only 1.4% asserted possessing advanced knowledge.

These technology-related characteristics of the sample align partially with the outcomes concerning prior experience with ChatGPT. Approximately 16.2% were unaware of its existence, 31.9% had an account but had never used it, and 51.9% had used it at some point. Among the students who had employed it, 22% utilized it as a support for assignments, 16.4% for information retrieval, 8% out of curiosity, and 5% for writing support. Regarding prior training in ChatGPT, 25.5% reported self-directed learning, 47.7% had not received any training, and 26.9% had undergone some form of training from university professors, either through specific classroom activities or various sessions aimed at gaining a deeper understanding of the tool’s advantages and proper usage.

The dimensions of the “perception and use”, illustrated in Figure 1, indicate that students in social disciplines view ChatGPT as an easily accessible tool ($\bar{x}=$ 5.4 on a scale ranging from 1 to 7). Other dimensions show average values around the level corresponding to the “indifferent” category on the Likert scale used (see Table 4), with responses displaying minimal dispersion.

Analyzing student characteristics (Table 5), no significant differences were observed okin ChatGPT perception based on gender, average grades, or prior familiarity with the tool. Two criteria, familiarity with video games and programming knowledge, were used to evaluate software proficiency.

Students with average or advanced video game expertise tended to rate the tool more positively across dimensions such as perceived usage, utility, attitude, and intention to use. Conversely, those with advanced programming skills assigned higher scores to the social influence dimension in ChatGPT use, with a significantly elevated average compared to other categories ($\bar{x}=$ 4.67, *p* < 0.01). Table 4 further illustrates that increased familiarity with ChatGPT corresponds to more favorable evaluations of the tool. Students with knowledge and prior usage of artificial intelligence tend to rate its use more positively ($\bar{x}=$ 3.68, *p* < 0.01), perceive greater utility ($\bar{x}=$ 5.72, *p* < 0.001), exhibit a more positive attitude ($\bar{x}=$ 3.52, *p* < 0.001), and express a stronger intention to use ($\bar{x}=$ 4.53, *p* < 0.001) compared to those unfamiliar with or yet to use it. Concerning the “main use of ChatGPT” category, students utilizing it for academic support or information retrieval provided higher ratings across all dimensions, excluding social influence and security, when contrasted with those who had not used it. Lastly, the received training appears to negatively influence the credibility of students who underwent classroom training, especially in intensive sessions. Additionally, they report experiencing less social pressure to use the tool compared to those without training.

In the second part of the analysis, extreme perceptions of students were isolated by initially identifying the most positive evaluations of ChatGPT usage and then incorporating student characteristics into the analyses. The same process was employed for negative evaluations. The findings are depicted through reticular graphs for each dimension, featuring only statistically significant associations (*p* < 0.05). As previously outlined, the node’s size indicates the response percentage, while the line thickness reflects the strength of the coincidence or relationship measured by the Haberman coefficient.

Network 1 reveals positive connections among the seven dimensions of Yilmaz’s scale. Particularly, “ease of use” stands out as a highly valued aspect of ChatGPT, strongly linked to increased “credibility” and an overwhelmingly positive “attitude” towards the tool. The dimension most closely associated with others is the “perception of its usage”. Although “perceived security” shows a less robust connection with the other dimensions, it still receives favorable ratings (Figure 2).

Network 2 shows how previous experience with ChatGPT emerges as a crucial factor strongly linked to four dimensions: “intention to use”, “perceived usefulness”, “attitude”, and “ease of use”. Students who have used it for academic support view ChatGPT as easy to use and beneficial. Meanwhile, those using it as a search tool exhibit a highly positive attitude toward its use, perceiving it as both user-friendly and secure. In the “security” dimension, females perceive the tool as more secure. Students employing ChatGPT for writing support not only maintain a positive attitude towards the tool but also express a clear intention to continue its use in the future.

Concerning students’ technological proficiency (evaluated through their familiarity with video games and programming), there is a link between advanced skills in video games and proficiency in programming. Students with advanced programming abilities also show a social influence and assign high “credibility” to the tool. Conversely, they align with students possessing advanced video game knowledge, indicating a stronger “intention to use” and a more positive evaluation of its “usefulness” and “attitude” toward ChatGPT. Students with average video game usage only display positive associations with the “easiness” dimension (Figure 3).

Regarding training with ChatGPT, there is a significant association between the lack of training and high credibility, as well as between self-directed training and an extremely positive attitude toward the tool. In essence, these two nodes are connected. Lastly, two students who failed the previous course showed a strong “intention to use the tool”, representing the only notable coincidence among academic grades.

After examining the coincidences based on negative Likert scale responses, a similar analysis of coincidences was conducted by grouping all negative responses (Network 3). In this context, it can be observed that the attitude toward the potential use of ChatGPT is associated with low “credibility” and an overall negative attitude towards the “use” of ChatGPT. Those who do not find the use of the tool easy only are related with those who have a negative perception, with no overlap between the other dimensions. The most critical ratings focus on “credibility” and “intention to use” (indicated by the size of the node), while “easiness” receives fewer negative responses (Figure 4).

Analyzing the relevant characteristics of students alongside the negative evaluation of dimensions (Network 4), the initial observation is the lack of any significant coincidence related to the “security” dimension, making it the only one isolated in the network among all analyzed dimensions. As indicated in Table 4, there do not appear to be discernible patterns explaining variations in the perception of security among students who, overall, seem rather indifferent in this regard.

It is intriguing to observe a kind of mirror effect between student characteristics coinciding with negative ratings of “perception” and “use” of ChatGPT compared to the results of more positive ratings. The category with the highest number of coincidences is that of students who have never used the tool: they perceive it as difficult, show a negative attitude towards its use, have no intention of using it in the future, and lack social influence to encourage its use. Among these students, those who were not previously aware of the tool stand out; most of them negatively assess a large part of the dimensions (attitude, usefulness, ease, intention, and influence). Additionally, those who, despite being aware of it, had not used it yet also display negative attitudes and intentions of use. Students who have used ChatGPT for writing give it low credibility, while those who have never used it declare not to be socially influenced to use the tool (Figure 5).

In terms of technological competencies, students with basic knowledge of video games maintained negative attitudes and intentions of use, as well as a poor perception of its utility. However, students with advanced programming knowledge coincided with those who did not find it easy to use ChatGPT. Concerning women, those who positively highlighted the tool’s security coincided with those who negatively assessed its credibility and intention of use. Additionally, significant coincidences were observed among students with an undeclared gender and a negative perception of utility. The type of education received at the university seems to generate rejection of ChatGPT usage, with significant coincidences in the dimensions of “credibility”, “intention of use”, “attitude”, “utility”, and “influence”. The types of grades do not seem to determine negative evaluations of the perception and use of ChatGPT.

This study aimed to analyze the education received during the current academic year (Network 5). Students who have learned to use ChatGPT on their own show coincidences in terms of behavior, social influence, and attitudes. Regarding the characteristics of these students, there are coincidences with usage for school tasks, male gender, advanced technological competencies, and, in terms of grades, these are students who did not pass the last evaluation (Figure 6).

The training for ChatGPT received at the university does not significantly affect its usage. Among the students who underwent training, many hold a negative perception of ChatGPT yet intend to use it. Within this group, a noticeable alignment is observed between women and high-performing students. Individuals who have not received any training in ChatGPT only value positively the “ease of use”.

## 5. Discussion

The survey results indicate a somewhat indifferent attitude towards ChatGPT among the surveyed students in the field of social sciences, contrasting with the significant interest observed among teachers and academics. This finding diverges somewhat from other studies, where students have displayed more favorable perceptions towards using this tool [72,75,89,96,105,109]. It aligns with research that reports neutral and less favorable student opinions when evaluating the tool’s usefulness, both within and outside the technology acceptance model (TAM) framework [79,93,110,111]. 

The reality that students hail from social science backgrounds, where there is a notably lesser presence of technology, may have adversely impacted their views and acceptance of this artificial intelligence tool. Unlike pure or natural sciences, which often provide straightforward “correct” or “incorrect” answers, the social sciences involve a higher degree of subjectivity and place greater emphasis on critical analysis. This inherent complexity in their fields could lead students to be more skeptical or critical of how artificial intelligence tools, which may seem rigid or overly simplistic, integrate into their nuanced academic processes. 

Regarding perceived usefulness, previous studies found that this dimension influences attitudes and intentions to use technology [11,14,16,75,77,101,112,113,114]. This study effectively confirmed the hypothesized relationship, as familiarity with video games and programming skills were found to have a significant association with a positive evaluation of ChatGPT. Participants who possessed greater knowledge and experience in these areas tended to appreciate the capabilities and potential of the AI tool more than those with limited exposure, suggesting that technical proficiency can influence perceptions and acceptance of new technologies. 

According to Liu and Zheng [24], credibility is a critical factor influencing people’s trust in technology, a finding supported by the responses obtained in this study. Yilmaz emphasizes the dimension of perceived social impact, stating that acceptance and technological adoption behaviors are socially influenced [72]. However, within the context of our study, this dimension does not appear to be a significant variable influencing the outcomes. We found that most students categorized their responses as “indifferent”, indicating a general lack of strong feelings or decisive opinions about the subject matter. This neutrality suggests that other factors may play a more critical role in shaping their perspectives, or that the dimension in question does not resonate strongly with their personal or academic experiences.

Research on the importance of learning and prior experience in technology acceptance revealed that users with more experience in using AI or chatbots may perceive ChatGPT as more useful and easier to use [45,101]. According to this study, students who are well-versed in video games and programming appear to have developed a more robust sense of familiarity, comfort, and confidence when interacting with AI-based systems. This seems to translate into more positive attitudes and intentions to use ChatGPT. Yilmaz [72] also found that students have favorable perceptions of ChatGPT in their educational experiences. However, there is a decreased level of confidence among students who have undergone training, particularly in sessions that involve several different exercises in an intensive format. 

Regarding gender, Cai et al.’s meta-analysis suggests that men have a more favorable attitude towards technology [115]. Similarly, Yilmaz [72] found gender differences in the perceived ease of use of ChatGPT, with men finding it easier to use. The study by Raman et al. [106] shows that students who were men favored compatibility, ease of use, and observability, while students who were women highlighted ease of use, compatibility, relative advantage, and trialability. More specifically, they suggest that men and women may have different expectations and experiences with technology. Men are more focused on functional aspects, while women are more attuned to operational and ethical implications, which could influence how ChatGPT is adopted and utilized in different learning environments. Under this assumption, they designed a study focusing on the role of gender, which revealed that students who were men found ChatGPT easy to explore and comprehend for their daily use. Students who were women found ChatGPT to have a greater advantage than other tools for day-to-day use and found it easy to understand, explore, and use. However, like men, women do not find ChatGPT easy to learn simply through observation. According to their findings, both genders found ChatGPT to be easy to use and understand, and that the ability to explore ChatGPT may be a key attribute influencing their usage and adoption intentions. The study of Bouzar et al. [116] found no significant gender difference in ChatGPT acceptance but variations in usage and concerns. Men reported longer usage times, while women reported higher usage frequency and greater apprehension about over-reliance on ChatGPT. Both genders found ChatGPT useful for educational purposes. 

The perception of ChatGPT as “male”, as identified in Wong’s study [117] may be a contributing factor to the observed gender differences in the acceptance of this technology. In our study, no gender differences were observed in the perception or use of ChatGPT, which is consistent with other studies based on the TAM [116] and another piece of research examining students’ perceptions from different perspectives [118]. This result also aligns with the literature review by Goswami and Dutta [119], who found that gender differences in technology use and acceptance do not emerge universally but in specific contexts. However, our sample is highly feminized, which could have affected our results.

Debates into gender differences in the acceptance of technology, both in general and in artificial intelligence, may yield complex and sometimes contradictory results. On one hand, it has been suggested that men might exhibit higher levels of comfort and positivity attitudes potentially due to greater exposure to technological fields such as engineering and computer science, which are traditionally male dominated. This exposure could lead to more familiarity and thus a higher level of acceptance. On the other hand, there is a growing body of research indicating that there are no significant gender differences. This could be attributed to the increasing democratization of technology access and education, bridging the gap between genders in tech-related fields. Additionally, societal shifts towards greater gender equality in educational and professional opportunities lead to similar levels of exposure and competency in technology, which could equalize perceptions and acceptance of AI across genders. Therefore, further research on the specific topic of acceptance model applied to IA and ChatGPT addressing gender differences is needed to answer this question.

Contrasting views and mixed results that we have reviewed in the literature suggest that university students’ attitudes towards ChatGPT and AI tools are complex and can vary significantly. This indicates a need for further research and understanding of different perspectives. While students find ChatGPT innovative, user-friendly, and compatible with their educational goals, ethical concerns regarding creativity, plagiarism, and academic integrity hinder its widespread acceptance. These barriers highlight the complex interplay between technological advancements, ethical considerations, and educational practices in the adoption of AI tools like ChatGPT among university students.

## 6. Conclusions

The main conclusions that emerge from this study can be summarized in the following key points:Students in social sciences exhibit notable indifference to ChatGPT, contrasting with the keen interest of the academic community. This may stem from a lower technological presence in social science disciplines and the survey’s timing in October and November of 2023, potentially when students were less acquainted with it.Perceived utility emerged as a key factor influencing attitudes and intentions toward technology use, supporting previous findings. Students with knowledge of video games and programming displayed statistically more positive attitudes towards ChatGPT.Credibility was confirmed as an influential factor in users’ confidence in this technology. The perceived dimension of social impact seemed less relevant for students in social sciences, suggesting variable dynamics in technological acceptance and adoption in different educational contexts.No significant gender disparities were observed in the perception or use of ChatGPT among social sciences students, supporting the idea that gender may not be a determining factor in all contexts, as suggested by previous studies.These conclusions provide valuable insights for designing training strategies that can enhance the acceptance of ChatGPT in social science education contexts, and above all, encourage responsible and appropriate use of this tool by emphasizing its strengths and weaknesses.

### 6.1. The Limitations of the Study

The primary limitations of this study are mostly related to the sample. The students who participated come from the social sciences, which may not be representative of the broader population or other academic disciplines, potentially affecting the generalizability of the findings. Conducting the survey in October and November 2023 may have influenced the results due to varying levels of exposure and familiarity with AI tools among students at that time. The novelty of ChatGPT might not have fully permeated the student body, leading to the indifference or unfamiliarity that we found. Additionally, the lower technological presence in social science disciplines might have skewed the results, as students from these fields may have less interaction with ChatGPT compared to their peers in more tech-oriented disciplines. Furthermore, the study relies on self-reported data, which can be subject to biases such as social desirability bias or inaccurate self-assessment. Lastly, the fact that there were more participants who were women raises questions about how this gender factor might have affected the results according to the literature discussed. Despite it not being clear whether gender differences matter when it comes to technology acceptance and use, we cannot exclude this from consideration.

Future studies should consider a longitudinal approach to assess changes in attitudes and perceptions over time as students become more familiar with ChatGPT. Additionally, expanding the study to include students from diverse disciplines and geographic locations would provide a more comprehensive understanding of the factors influencing ChatGPT acceptance.

### 6.2. Implications of the Study

The findings presented in this work indicate a need for tailored educational strategies that highlight both the strengths and limitations of technology to ensure informed and critical use. Higher education institutions have the responsibility to lead these initiatives. Moreover, higher education teachers must be aware of how students are using tools like ChatGPT and similar AI technologies. Failing to do so could result in falling behind in technological advancements and misunderstanding students’ use of AI, which could impact evaluations and grading. To address the challenges posed by AI, it is essential to consider various strategies and initiatives. These should align with the factors influencing technology acceptance and use, as outlined in the following figure (Figure 7):

It is not about advocating for or against ChatGPT, but understanding how students are reacting to the widespread adoption of this AI and what factors explain their attitudes. For instance, we have found that perceived utility is a key factor that leads students to use ChatGPT. By highlighting the utility of other academic alternatives or complementary tools to ChatGPT, we can provide more choices to students. Credibility was another important factor, and it is therefore essential to help students develop realistic expectations about AI. Again, it is not about demonizing or romanticizing what ChatGPT can or cannot do, but rather teaching students the importance of verifying information and being analytical, regardless of the tool, resource, or instrument they are using. To this end, it is also essential that teachers introduce ChatGPT into the classroom and critically discuss both correct and incorrect feedback with their students.

Given the positive correlation between knowledge of video games, programming, and favorable attitudes toward ChatGPT that we found, integrating introductory courses on these topics could improve overall competency in using AI tools. ChatGPT has the potential to enhance accessibility by providing personalized learning experiences and offering supplementary support to students with or without special needs. However, it is essential to address potential hindrances such as dependence on technology and accessibility due to economic resources, i.e., a possible gap between those students that can afford the Pro versions of AI tools and those who cannot. In short, a careful integration of ChatGPT into educational settings need to address many factors to ensure that AI complements and helps rather than replaces human interaction and creates inequality. Educators can leverage its benefits while mitigating its drawbacks, thus promoting an inclusive and fair learning environment for all students. For this reason, we need to explore these issues, including the teachers’ perceptions and behaviors that we did not cover in this study.

## Figures and Tables

**Figure 1 behavsci-14-00612-f001:**
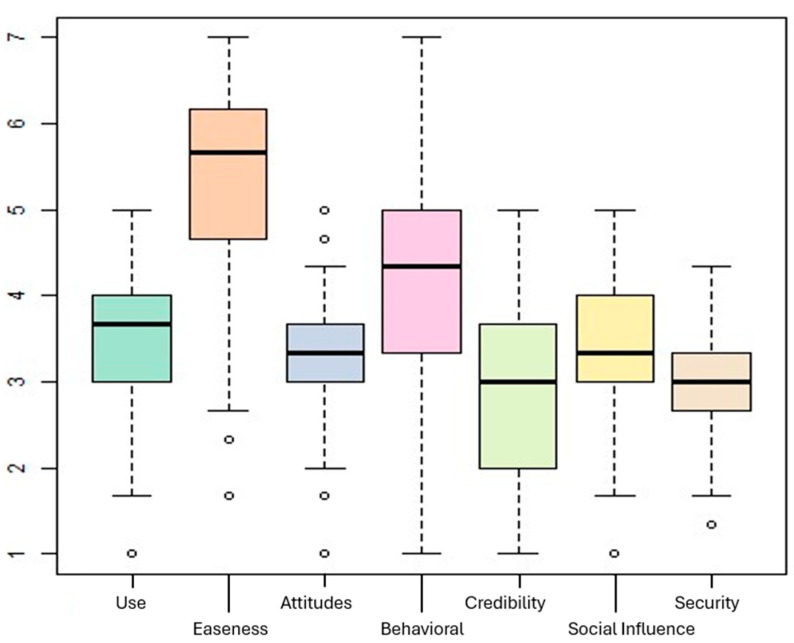
Dimensions of perception and usage of ChatGPT.

**Figure 2 behavsci-14-00612-f002:**
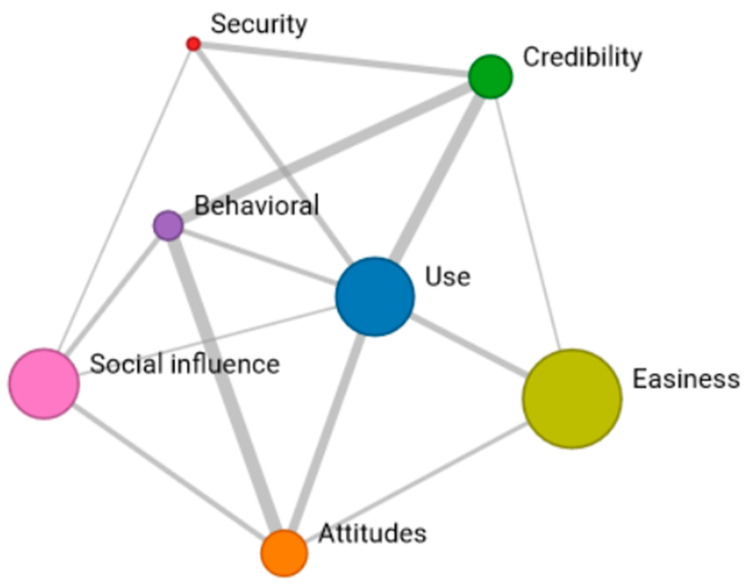
Network 1: Reticular coincidences in the “very positive” response of the seven dimensions.

**Figure 3 behavsci-14-00612-f003:**
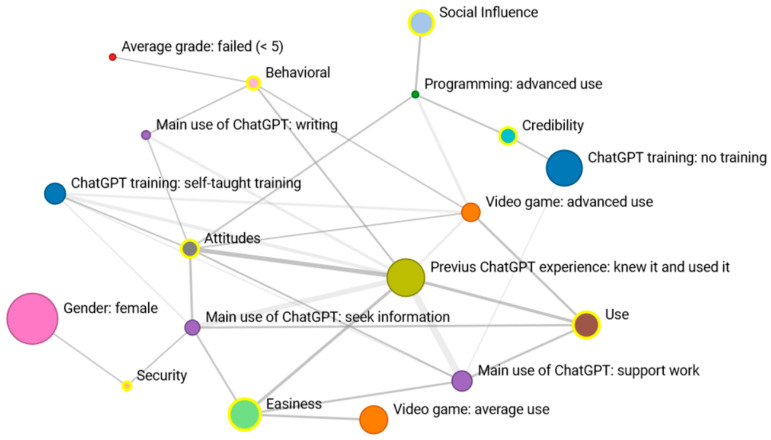
Network 2: Reticular coincidences in the “Very positive” response of the 7 dimensions according to student characteristics.

**Figure 4 behavsci-14-00612-f004:**
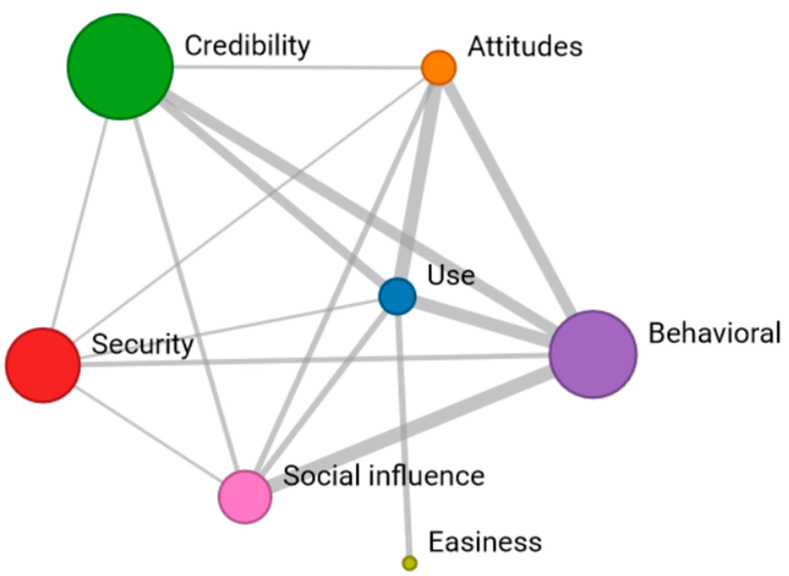
Network 3: Reticular coincidences in the “very negative” response of the 7 dimensions.

**Figure 5 behavsci-14-00612-f005:**
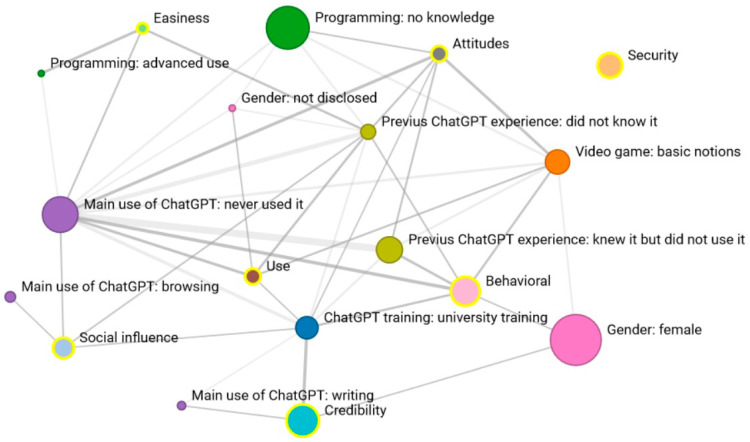
Network 4: Reticular coincidences in the “very negative” response of the 7 dimensions according to student characteristics.

**Figure 6 behavsci-14-00612-f006:**
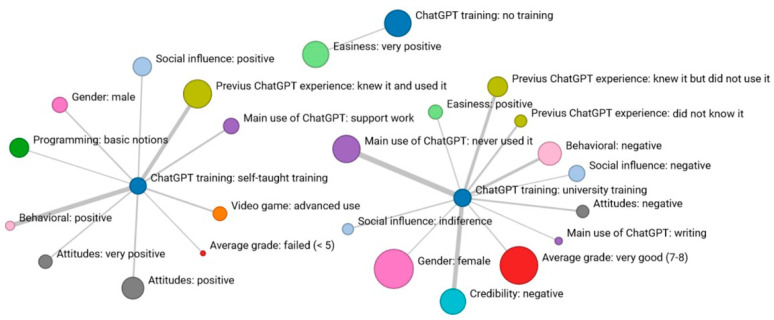
Network 5: Significant coincidences related to the type of training in ChatGPT.

**Figure 7 behavsci-14-00612-f007:**
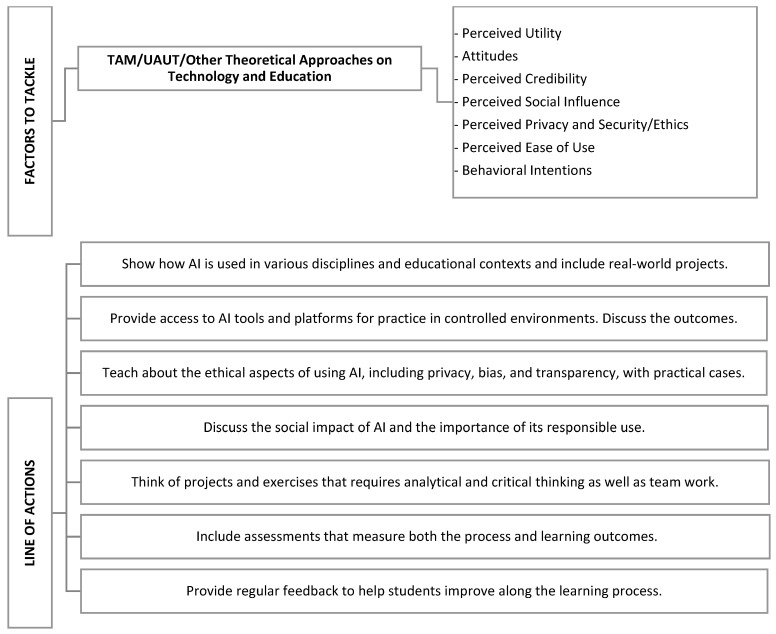
Lines of actions for effective programs.

**Table 1 behavsci-14-00612-t001:** Student characteristics.

Variable	Category	N = 216
Age	17–19	70% (150)
	20–22	22% (48)
	>23	7.7% (15)
Gender	Male	25.8% (55)
	Female	72.3% (154)
	No reply/others	1.9% (4)
Grade	Criminology (1°)	58.9% (126)
	Sociology (1° and 4°)	27.1% (58)
	Social education (2°)	14% (30)

**Table 2 behavsci-14-00612-t002:** Dimensions of the questionnaire on attitudes toward ChatGPT by Yilmaz [72].

(1) Perceived Utility	This refers to the degree to which a user believes that using ChatGPT will enhance their performance or productivity in a specific task. For instance, students might perceive that using ChatGPT can help them better understand complex topics or complete their assignments more efficiently.
(2) Attitudes Towards the use of ChatGPT	This factor encompasses the user’s overall affective reactions to using ChatGPT. It includes positive or negative feelings about the use of the technology, which are influenced by their experiences and perceptions of its usefulness and ease of use.
(3) Perceived Credibility	This refers to the extent to which users believe that ChatGPT provides accurate, reliable, and trustworthy information. Credibility can be influenced by factors such as the perceived expertise of the source and the consistency of the information provided.
(4) Perceived Social Influence	This factor involves the degree to which users perceive that important others (such as friends, colleagues, or instructors) believe they should use ChatGPT. Social influence can affect user attitudes and behaviors towards adopting new technologies.
(5) Perceived Privacy and Security	This refers to the degree to which users feel confident that their data and interactions with ChatGPT are secure and that their privacy is protected. Concerns about data breaches or misuse of personal information can significantly impact the acceptance of the technology.
(6) Perceived Ease of Use	This factor is about the degree to which a user believes that using ChatGPT will be free of effort. It includes perceptions of how easy it is to learn and use the system. If users find ChatGPT intuitive and user-friendly, they are more likely to adopt it.
(7) Behavioral Intention to use ChatGPT	This refers to the user’s intention to use ChatGPT in the future. Behavioral intention is influenced by attitudes towards use, perceived usefulness, perceived ease of use, and other factors. A strong behavioral intention typically predicts actual usage behavior.

**Table 3 behavsci-14-00612-t003:** Operationalization of ChatGPT perception and usage measurement.

Dimension	Items	Response Range
Perceived Utility	3	1. Strongly disagree; 2. disagree; 3. neutral; 4. agree; 5. strongly agree
Attitudes Towards ChatGPT Usage	3	
Perceived Credibility	3	
Perceived Social Influence	3	
Perceived Privacy and Security	3	
Perceived Ease of Use	3	1. Very difficult; 2. difficult; 3. somewhat difficult; 4. neutral; 5. somewhat easy; 6. easy; 7. very easy.
Behavioral Intention to Use ChatGPT	3	1. Very improbable; 2. improbable; 3. somewhat probable; 4. neutral; 5. somewhat probable; 6. probable; 7. very probable.

**Table 4 behavsci-14-00612-t004:** Frequency table of the dimensions of the index.

	Categories Grouped According to Likert Scale				Statistics			
	Negative	Indifferent	Positive	Very Positive	Total	N	Mean	SD
Use	16.7%	12.0%	38.0%	33.3%	100%	216	2.9	1.054
Attitudes	16.0%	25.0%	39.6%	19.3%	100%	212	2.6	0.973
Credibility	43.9%	15.4%	22.9%	17.8%	100%	214	2.1	1.168
Social Influence	23.6%	14.2%	32.1%	30.2%	100%	212	2.7	1.138
Security	31.8%	41.6%	22.4%	4.2%	100%	214	2.0	0.845
Easiness	8.7%	24.6%	19.5%	47.2%	100%	195	3.1	1.034
Behavioral	41.1%	34.9%	10.9%	13.0%	100%	192	2.0	1.022

**Table 5 behavsci-14-00612-t005:** Mean values of perception and usage of ChatGPT according to student characteristics.

		Use	Easiness	Attitudes	Behavioral	Credibility	Influence	Security
	Mean	3.50	5.40	3.29	4.11	2.89	3.35	2.95
	Standard deviation	0.71	1.08	0.68	1.44	0.89	0.76	0.49
Gender	Male	3.64	5.27	3.40	4.43	3.03	3.33	2.92
	Female	3.46	5.44	3.25	4.01	2.83	3.36	2.99
	Not disclosed	3.00	5.00	3.17	3.42	3.25	3.08	2.83
Grades	Fail (<5)	3.83	5.33	3.67	6.00	2.17	3.67	2.67
	Pass (5–6)	3.69	5.51	3.35	4.41	3.17	3.61	2.93
	Notable (7–8)	3.47	5.41	3.28	4.04	2.84	3.34	2.99
	Outstanding (9–10)	3.47	5.24	3.23	3.95	2.95	3.21	2.87
Video Game Knowledge	None	3.54	5.56	3.36	4.33	3.15	3.46	2.95
	Basic	3.27 **	5.05 **	3.01 **	3.75 *	2.75	3.34	2.97
	Average use	3.51	5.65 **	3.40	4.11	2.85	3.29	3.00
	Advanced use	3.78 **	5.39	3.50 **	4.53 *	3.00	3.39	2.89
Programming Knowledge	None	3.48	5.32	3.20	4.08	2.87	3.34 **	2.95
	Basic	3.46	5.67	3.39	4.11	2.89	3.28 **	2.94
	Average use	3.63	5.23	3.44	4.33	2.88	3.47	3.10
	Advanced use	4.22	4.33	4.00	4.89	3.89	4.67 **	3.11
Previous Experience with ChatGPT	Was aware of it and used it	3.68 **	5.72 ***	3.52 ***	4.53 ***	2.98	3.42	2.93
	Was aware of it but did not use it	3.35	5.16	3.05	3.68	2.84	3.30	2.95
	Was not aware of it	3.23	4.81	3.02	3.67	2.73	3.22	3.06
Main Use of ChatGPT	Support in assignments	3.74 **	5.81 **	3.47 **	4.52 **	3.16 **	3.49	2.89
	Information retrieval	3.78 **	5.73 **	3.65 **	4.99 **	3.06 **	3.50	3.04
	Curiosity	3.41	5.43	3.47	3.61 **	2.94	3.22	2.82
	Writing	3.52	5.73	3.42	4.55	2.18 *	3.33	2.85
	Never used it	3.31 **	5.05 ***	3.04 **	3.67 **	2.80 *	3.27	2.99
Type of ChatGPT Training	None	3.57	5.22	3.31	4.20	3.36	3.48 *	3.02
	One session	3.44	5.58	3.24	3.94	2.49 ***	3.18 *	2.93
	A few sessions with different exercises	3.54	5.28	3.45	4.56	2.86 *	3.56	2.86

Note: Levels of statistical significance for the difference of means: * *p* < 0.05, ** *p* < 0.01, *** *p* < 0.001.

## Data Availability

The data presented in this study are available on request from the corresponding author.

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
