# Peer review of "Training and Technology Acceptance of ChatGPT in University Students of Social Sciences: A Netcoincidental Analysis"

_behavsci, 2024, doi:10.3390/bs14070612_

Round 1

Reviewer 1 Report

Comments and Suggestions for Authors

1. Does the article address a topic of current concern about the aims of the journal?

Yes, the article addresses a topic of current concern, particularly the integration of Chat GPT in educational settings and its acceptance among university students. This is highly relevant to the journal's aims, which seem to focus on behavioural sciences and the impact of technology on them.

2. Does the abstract give a clear account of the scope of the paper?

The abstract provides a clear account of the paper's scope.

3. Do the keywords adequately reflect the paper?

They encompass the main themes and methodological approaches used in the study, making the article easily discoverable for interested readers.

4. Does the article consider relevant contemporary literature in the area?

The article appears to consider relevant contemporary literature, as evidenced by the numerous citations discussing the use of Chat GPT in various educational contexts and the implications for teaching and learning.

5. If the article concerns research activity, has a sound methodology been employed and described?

The research employs the Technology Acceptance Model (TAM) and Reticular Analysis of Coincidences (RAC) to analyse survey data. The approach is well-described, and the chosen methods are appropriate for the research questions.

I was wondering whether some qualitative data were collected for further analysis. Adding qualitative data to complement the quantitative findings could provide a richer understanding of students’ perceptions and experiences with Chat GPT.

In addition, the study mentions the impact of training on perceptions. It could be helpful to explore the characteristics of effective training programs and how they might be designed to maximise positive outcomes.

6. Does the article clearly distinguish between opinion and empirical evidence?

The survey results and subsequent statistical analyses are presented objectively, with interpretations based on the data. Discussions on the implications of findings are separate from the presentation of empirical results, maintaining a clear line between data-driven conclusions and the authors' perspectives.

The authors could consider using subheadings or explicit statements to separate these elements within the text.

7. Does the article contribute to a critical understanding of the issues?

The study contributes to a critical understanding of how AI tools like Chat GPT are perceived and utilised in educational settings. It highlights essential factors such as prior experience, training, and perceived utility, offering insights that can inform future educational strategies and policies.

If possible, the authors could provide more detailed recommendations for educators and policymakers on implementing the findings in practice to enhance the integration of Chat GPT in education. Additionally, as listed below, two potential directions could be explored further.

(1) Teachers' perspectives: Including instructors' perspectives on the use of Chat GPT could offer additional insights into its integration into educational practices.

(2) For accessibility and inclusion: The study could consider how Chat GPT affects students with different needs, including those with disabilities, and how it might enhance or hinder accessibility in education.

8. Does the article reference citations and quotations according to the submission guidelines?

References and citations are presented and adhere to the journal's submission guidelines.

Comments on the Quality of English Language

The English in the article is generally clear and coherent. The sentences are well-structured, and the ideas are presented in a logical sequence. While the article is written for an academic audience, it is essential to ensure that complex concepts are explained in a way accessible to readers from various backgrounds. Using more straightforward sentences or breaking down complex ideas could improve readability. The above-mentioned is just my personal opinion. : )

Author Response

Thank you very much for your positive feedback on our manuscript. We have improved each section to provide a enhanced conceptual clarity, improved methodological consistency, and superior writing quality, and to address all doubts and suggestions raised by the reviewers. We greatly appreciate these suggestions. In total, the changes and improvements made resulted in the addition of around 2000 new words, one new figure, one new table, and four new references.

In attachments, we provide a point-by-point summary of our response to your comments. These improvements are clearly highlighted in red in the attached revised manuscript. These modifications have substantially enhanced our work, and we are very grateful for your valuable input.

Reviewer 2 Report

Comments and Suggestions for Authors

The content of the article is well-contextualized within the theoretical and empirical background. The research design and methods are clearly articulated in the article. The discussion of findings is coherent and balanced. The article provides detailed descriptions of the statistical analyses performed, including the use of reticular graphs to highlight significant associations among variables. Moreover, the article is adequately referenced, citing a wide range of sources to support its claims.

Some minor suggestions:

Research Questions and Hypotheses: While the article addresses gaps in the literature, explicitly stating the research questions and hypotheses at the beginning of the methodology section would improve clarity. Currently, these elements are implied but not directly stated.

Methods: The methods section provides a thorough description of the survey and analytical techniques used. However, a more detailed explanation of the Reticular Analysis of Coincidences (RAC) and how it applies to the study might be beneficial for readers unfamiliar with this method.

Discussion of Findings: The discussion integrates the findings well with existing literature, but it could benefit from a deeper analysis of the implications of these findings. Specifically, exploring the practical applications of the results in educational settings and providing more concrete recommendations would strengthen the argument.

Future Research Directions: Expanding the discussion on limitations and suggesting detailed areas for future research would provide a clearer path forward for further investigation.

Overall, the article makes a valuable contribution to understanding the acceptance and usage of Chat GPT among social sciences students. It is a well-conducted study with meaningful implications for educational practices, despite areas needing improvement in clarity and engagement with recent literature.

Author Response

(The authors gave the same response as above.)

Reviewer 3 Report

Comments and Suggestions for Authors

This study transitions directly from the introduction to the Materials and Methods section, thereby missing a crucial element: the literature review. A robust literature review helps identify the research gap and clarifies the study's contribution to the field, establishing a strong foundation for the subsequent analysis and discussion.

literature review is missing.

literature review should be structured to cover several key areas that provide a comprehensive foundation for the research. Here are the suggested sections:

1. Introduction to Technology Acceptance Models (TAM)

2. Chat GPT and Artificial Intelligence in Education

3. Factors Influencing Technology Acceptance Among University Students

you have to discuss the below factors

Perceived Utility

Attitudes towards Chat GPT Usage

Perceived Credibility

Perceived Social Influence

Perceived Privacy and Security

4. Training Strategies for Enhancing Technology Acceptance

5. Implications of Technology Acceptance in Educational Contexts

at the end add paragraph summary of key findings from the literature Identification of research gaps.

In addition, the implication section is missing

Moreover, Limitation and direction for future research is missing

Author Response

(The authors gave the same response as above.)

Round 2

Reviewer 3 Report

Comments and Suggestions for Authors

as I wrote before the structure of paper must to be

1. introduction

2. literature review

no subsection under introduction

all subsection must to be under the literature review

literature review should be structured to cover several key areas that provide a comprehensive foundation for the research. Here are the suggested sections:

2. Literture Review

2.1 Introduction to Technology Acceptance Models (TAM)

2.2 Chat GPT and Artificial Intelligence in Education

2.3  Factors Influencing Technology Acceptance Among University Students

you have to discuss the below factors

Perceived Utility

Attitudes towards Chat GPT Usage

Perceived Credibility

Perceived Social Influence

Perceived Privacy and Security

2.4 Training Strategies for Enhancing Technology Acceptance

2.5 Implications of Technology Acceptance in Educational Contexts

2.6 Related Works

at the end add paragraph summary of key findings from the literature Identification of research gaps.

Author Response

Dear Editors and Reviewers,

Thank you very much for your time and dedication. We appreciate the agreement of R1 and R2 with the requested changes. Below are the changes made to satisfy the requests of R3:

A new section titled "Literature Review" has been created, and three subsections have been added in the order suggested by the reviewer. This involved adding new information to the introduction and moving some paragraphs from the introduction to the new sections, as well as reviewing the literature and writing new paragraphs for the suggested theoretical review.

Most of the studies introduced were published in late May and June and did not exist when we conducted this work in October. We even searched through lesser-known academic engines and preprints, increasing the references from 87 to 123, some of which have also been introduced in other sections. In total, almost 2000 words have been added.

Despite significantly increasing the number of references compared to the initial ones, we believe that there is not yet a large enough body of knowledge to create specific subsections on the dimensions of the TAM model, specifically on the items of the tool we have used. There is a lot of literature on this model, but only since the arrival of Chat GPT has it begun to be applied to this tool, and students are a specific sample type. Therefore, the number of studies is limited. Additionally, many investigations have not applied the TAM model but have addressed interesting factors that explain acceptance and use. For this reason, the information requested by R3 has been addressed in sections 2.2 and 2.3 along with other factors within and outside this theoretical model.

We ask the reviewer not to see our decision as an intention or will to contravene their indications, but rather that we have done what we could with the quality of the paper and the argumentative thread in mind.

Two studies published in June focus on the gender variable, and therefore we have introduced them in the discussion, even though this change was not requested.

The color code used is as follows:

- Red: changes made in the first review

- Purple: changes made in the second review specifically addressing R3.

We also took the opportunity to correct a small error in the surname of one of the co-authors and to indicate who should be the corresponding author.

Round 3

Reviewer 3 Report

Comments and Suggestions for Authors

The revised paper is better than before.